

**The Role of Chemical Boundary Conditions in Simulating Summer Ozone and**
**Cross-Boundary Transport over China**

Yunsong Du[1,2], Fumo Yang[1], Sijia Lou[3], Baolei Lyu[4], Ran Huang[5], Guangming Shi[1],
Yongtao Hu[6], Yan Jiang[7], Nan Wang[1*]
[1]College of carbon Neutrality Future Technology, Sichuan University, Chengdu 610065, China
[2]Department of Environmental Science and Engineering, Sichuan University, Chengdu 610065,
China
[3]School of Atmospheric Sciences, Nanjing University, Nanjing 210023, China
[4]Huayun Sounding Meteorological Technology Co. Ltd., Beijing 100081, China
[5]Hangzhou AiMa Technologies, Hangzhou, Zhejiang 311121, China
[6]School of Civil and Environmental Engineering, Georgia Institute of Technology, Atlanta,
Georgia 30332, USA
[7] Sichuan Eco-environment Monitoring Station, Chengdu 610091, China
**\*Correspondence:** Nan WANG (nan.wang@scu.edu.cn)



**Key Points**

1. We systematically evaluated the impacts of chemical boundary conditions (static vs. dynamic) on regional $O_3$ simulations over China.

2. Chemical boundary conditions strongly modulate $O_3$ simulations via cross-boundary transport in both horizontal and vertical directions.

3. Synoptic circulation dynamically amplifies the impacts of chemical boundary conditions on regional $O_3$ levels.





**Abstract**
Regional chemical transport models are vital for diagnosing and forecasting
tropospheric ozone ($O_3$) pollution. However, their accuracy is often limited by the
simplified treatment of chemical boundary conditions (CBCs). This study provides a
comprehensive evaluation of how different CBCs influence regional $O_3$ simulations
over China using the WRF–CMAQ model. Four CBCs scenarios were assessed: a static
BASE profile representing climatological conditions and three dynamic scenarios
derived from H-CMAQ, GEOS-Chem, and CESM2.2. Model results were validated
with surface networks, ozonesonde profiles, and satellite $O_3$ columns. The BASE
scenario underestimated the average maximum daily 8-hour $O_3$ (avg-O3MDA8) and its
90th percentile by −5.7% and −13.1%, respectively, while dynamic CBCs substantially
improved the accuracy. GEOS-Chem achieved the lowest bias (−0.3%) and highest
agreement (0.85 and 0.83) for avg-O3MDA8 and its 90th percentile. H-CMAQ
performed best in high-elevation northwestern regions, and CESM2.2 excelled in
southern and southwestern areas. Vertically, all CBCs reasonably matched observations
within the troposphere, but elevated lower-stratosphere biases were identified in BASE,
H-CMAQ, and CESM2.2. A case study contrasting cyclone-scavenging and post-trough
accumulation phases revealed that dynamic CBCs enhance cross-boundary transport
efficiency, raising $O_3$ by 10–20% over eastern China through combined continental and
stratospheric inflows. These results underscore the crucial role of synoptic circulation–
driven transboundary transport in shaping regional $O_3$ concentrations and demonstrate
the importance of realistic, time-varying CBCs for improving regional $O_3$ simulations,
air quality forecasting, and transboundary pollution management in China.

**Key words**: $O_3$ simulation; cross-boundary transport; chemical boundary condition;
chemical transport model





## 1 Introduction

Ozone ($O_3$) pollution is a critical environmental issue with profound implications for air quality (Malley et al., 2017; Chiu et al., 2023). As a secondary pollutant, tropospheric $O_3$ is mainly formed through photochemical reactions involving precursors such as nitrogen oxides ($NO_x$) and volatile organic compounds (VOCs) under sunlight. Elevated $O_3$ concentrations pose severe risks to public health, contributing to respiratory diseases and premature mortality, while also damaging ecosystems and suppressing agricultural productivity (WHO, 2016; Wang et al., 2017; Zhang et al., 2019a). In addition, as a highly reactive oxidant, tropospheric $O_3$ regulates the atmospheric lifetime of numerous reactive trace gases by governing their chemical transformations (Jacob, 2003).

$O_3$ pollution is currently one of the most pressing environmental challenges faced globally. In many Western countries, stringent air pollution controls implemented since the last century have led to stabilization or even declines in $O_3$ concentrations (Monks et al., 2015; Tarasick et al., 2019). Over the past decades, China have experienced frequent high-ozone episodes, drawing increasing attention from both the scientific community and policymakers. Even though, China has only more recently undertaken aggressive air quality improvement measures, most notably through the Air Pollution Prevention and Control Action Plan launched in 2012, which mandated substantial reductions in nitrogen oxide emissions. Despite these efforts, $O_3$ concentrations in China have not shown a sustained decline; instead, they have continued to rise in major urban agglomerations such as the North China Plain and the Yangtze River Delta (Zhang et al., 2019b; Lu et al., 2020; Wang et al., 2020; Wang et al., 2022). Nevertheless, the formation and distribution of $O_3$ are governed by the interplay of precursor emissions, meteorology, and transport processes. Variations in the magnitude and composition of anthropogenic and natural $NO_x$ and VOC emissions shape the chemical regime for $O_3$ production and loss. Meteorological conditions (e.g., temperature, solar radiation, humidity, boundary layer dynamics, and circulation patterns) further modulate photochemical reaction rates, vertical mixing, and horizontal transport, while surface characteristics and complex topography can influence local stagnation and recirculation. Together with regional and transboundary transport, as well as inflow from the free troposphere and occasional stratosphere–troposphere exchange, these processes determine background $O_3$ levels and lead to strong spatial and seasonal heterogeneity in $O_3$ pollution (Monks et al., 2015; Lu et al., 2018).Regional chemical transport models (CTMs) are essential tools for predicting and diagnosing air pollution, particularly photochemical $O_3$ pollution. Unlike global models, which emphasize large-scale atmospheric processes at coarse spatial resolutions, regional models such as the Community Multiscale Air Quality (CMAQ) and the Weather Research and Forecasting model with Chemistry (WRF-Chem) resolve chemical and physical processes at finer spatial and temporal scales (Byun and Schere, 2006; Grell et al., 2005). This capability enables them to capture the complex interactions among local emissions, meteorology, and topography that govern the formation, transport, and dispersion of $O_3$ and its precursors. However, the reliability of CTM-based $O_3$ simulations ultimately depends on the accuracy and consistency of the meteorological fields, emission inputs, and



chemical boundary conditions that define the model environment (Hogrefe et al., 2018;
Solazzo et al., 2012).
Over the past decade, CTMs have become central to air quality forecasting, scientific
research, environmental assessment and policy evaluation (Yahya et al., 2015; Bai et
al., 2018; Wang et al., 2021b; Gao et al., 2024). Their flexible domain configurations
allow targeted simulations over regions with intense emissions or complex terrain, such
as urban agglomerations and mountainous areas (Wang et al., 2019; Mao et al., 2022a;
Dou et al., 2024). Besides, incorporating high-resolution emission inventories and
region-specific meteorological inputs further enhances their accuracy. Numerous
applications have demonstrated their scientific and practical value: Zhang et al. (2019b)
used WRF-CMAQ to disentangle the relative roles of anthropogenic emissions and
meteorology in $PM_{2.5}$ variability, while Mao et al. (2022a) reproduced multi-pollutant
trends across China between 2013 and 2019, validating CMAQ's long-term
performance. Wang et al. (2024) applied CMAQ to assess regional $O_3$ responses during
heatwaves, highlighting the strong sensitivity of $O_3$ formation to both emissions and
meteorological drivers. Collectively, these applications underscore the indispensable
role of regional CTMs in advancing mechanistic understanding of air pollution and in
guiding effective clean-air strategies (Yahya et al., 2015; Lei et al., 2023; Dou et al.,
2024; Geng et al., 2024).
Building on this foundation, substantial efforts have focused on improving the
performance and reliability of regional CTMs. A major area of optimization lies in
chemical mechanisms: updated frameworks such as Carbon Bond 6 (CB6) and SAPRC-
11 enhance model fidelity in representing $O_3$ formation pathways and secondary organic
aerosol production under diverse atmospheric conditions (Yarwood et al., 2010; Carter
and Heo, 2013). Parallel improvements in meteorological simulations—through
techniques such as four-dimensional data assimilation (FDDA) in WRF and the
incorporation of high-resolution land-use datasets (e.g., MODIS, NLCD)—have
sharpened the representation of surface temperature, planetary boundary layer height,
and wind fields (Mallard et al., 2018; Campbell et al., 2019; Godowitch et al., 2015;
Wang et al., 2021a; Siewert and Kroszczynski, 2023). Meanwhile, advances in
anthropogenic and biogenic emission inventories, including the Multi-resolution
Emission Inventory for China (MEIC) and the U.S. National Emissions Inventory (NEI),
now provide finer spatial and temporal detail, capturing sector-specific variability and
reducing input uncertainty (Li et al., 2017a; Zheng et al., 2021; Foley et al., 2023; Geng
et al., 2024). Together, these continuous advancements have considerably strengthened
the capacity of regional CTMs to support both scientific inquiry and evidence-based
policy-making.
Despite substantial advances in regional chemical transport models (CTMs),
comparatively little attention has been devoted to chemical boundary conditions
(CBCs), even though they critically influence model accuracy. CBCs specify the
concentrations of air pollutants at the lateral and vertical boundaries of the simulation
domain, thereby constraining internal chemical evolution and pollutant transport
(Goldberg et al., 2015; Hogrefe et al., 2018). Accurate CBCs are essential for capturing
the impact of long-range pollutant transport and representing background



concentrations, both of which strongly shape regional air quality. For regional O₃, these
boundary-driven background levels can modulate the effectiveness of local emission
controls, alter the chemical sensitivity regime, and partly determine the spatial gradients
between upwind and downwind areas. In regions strongly influenced by continental
outflow, stratosphere–troposphere exchange, or marine inflow, poorly specified CBCs
may therefore lead to systematic biases in simulated O₃ distributions(Zhu et al., 2024;
Goldberg et al., 2015; Hogrefe et al., 2018). Oversimplified treatments—such as
prescribing fixed background values or climatological means—can introduce
substantial biases, resulting in misrepresentation of pollutant levels and misleading
evaluations of source contributions, policy effectiveness, and health risks (Yahya et al.,
2015; Hogrefe et al., 2018). Indeed, sensitivity studies show that uncertainties in CBCs
can alter simulated O₃ by several parts per billion, with particularly pronounced effects
in downwind and coastal regions influenced by transboundary transport (Hogrefe et al.,
2018; Jerrett et al., 2005) .
In China, few studies have systematically assessed the role of chemical boundary
conditions in influencing model performance or pollutant attribution across different
geographical regions (Zhu et al., 2024). This represents a critical gap, as the spatial
heterogeneity of transboundary influences—from continental transport in the west to
marine outflow in the east—could lead to regionally differentiated impacts on pollutant
concentrations and control policy outcomes. For example, western and northern China
may be more strongly affected by inflow of polluted air masses from upwind continental
source regions, while eastern coastal areas can be influenced by recirculation and clean
or polluted marine air, leading to distinct baseline O₃ levels and vertical structures.
Without an explicit assessment of CBCs across these contrasting regimes, regional
CTM applications may under- or overestimate O₃ burdens and misattribute observed
spatial patterns to local emissions or meteorology alone (Solazzo et al., 2012; Ni et al.,
2018; Sahu et al., 2021; Mao et al., 2022b; Shen et al., 2024).Therefore, a
comprehensive evaluation of the role of chemical boundary conditions in regional CTM
applications is urgently needed to enhance model reliability, reduce forecast uncertainty,
and support the formulation of more effective O₃ mitigation strategies.
Herein, we used outputs from three global chemical transport models to provide
downscaled CBCs for the regional CMAQ model and systematically evaluated the
impact of including versus omitting CBCs on O₃ simulations. Surface observations,
ozonesonde profiles, and satellite data were used to assess model performance across
China. We also examine the mechanisms by which CBCs influence O₃, including their
regulation of background concentrations and propagation of transboundary pollutants
into the domain. This study advances understanding of CBCs in regional air quality
modeling and provides a foundation for more accurate high-resolution O₃ forecasts and
improved air quality management strategies. By explicitly contrasting simulations with
and without chemically consistent CBCs, while keeping emissions and meteorology
unchanged, this study isolates the contribution of boundary conditions from other key
drivers of O₃ variability. The resulting diagnostics provide a clearer physical
interpretation of how CBCs interact with regional emissions and meteorological fields
to shape O₃ distributions over China.





**2 Data and Method**

**2.1 Modelling Configuration**

In this study, $O_3$ concentrations during July–August 2019 were simulated using the WRF-CMAQ modeling system. The Weather Research and Forecasting (WRF, https://www.mmm.ucar.edu/models/wrf) model version 3.9.1 was used to generate meteorological fields, which were then provided as inputs to drive the Community Multiscale Air Quality (CMAQ) model version 5.3.3 (http://www.epa.gov/cmaq). CMAQ solves the governing physical and chemical equations to simulate the three-dimensional distribution of air pollutants. The simulations were conducted at a horizontal resolution of 36 km, covering the entire Chinese mainland and surrounding regions to ensure adequate representation of regional transport processes. (see Fig. 1). The meteorological initial and boundary conditions were derived from the ERA5 reanalysis dataset (0.25° × 0.25° resolution), provided by the Copernicus Climate Change Service via the Climate Data Store (CDS) (Hersbach et al., 2023) . Anthropogenic emissions over China were obtained from the Multi-resolution Emission Inventory for China (MEIC v1.4) for the year 2019 (Li et al., 2017a), which provides sector-based emissions mapped to CMAQ species (http://meicmodel.org, last accessed: January 1, 2024). For regions outside China, the MIX v1.1 inventory was used, which is also developed by Tsinghua University (Qiang Zhang) with input from Asia Center for Air Pollution Research (Jun-ichi Kurokawa and Toshimasa Ohara), Konkuk University (Jung-Hun Woo), Argonne National Laboratory (Zifeng Lu and David Streets), and Peking University (Yu Song) (Li et al., 2017b) and includes regionalized emissions for East Asia. The biogenic emissions were estimated using the inline Biogenic Emission Inventory System (BEIS3) embedded within CMAQ which dynamically calculates emissions based on land use, vegetation type, and meteorological conditions online. The gas-phase chemistry was represented using the SAPRC07TC mechanism, while aerosol processes were simulated using the AERO6 module.

In order to assess the influence of CBCs on $O_3$, four different CBC scenarios were designed and applied as inputs to the CMAQ BCON (boundary condition) module. The first scenario, referred to as BASE, employs a spatially uniform and temporally constant boundary condition derived from the built-in ASCII vertical profiles in CMAQ. These profiles were extracted from a hemispheric CMAQv5.3 beta2 simulation for the year 2016, representing annual mean concentrations at the grid cell nearest to (37N, -157W), which is over the ocean in the central North Pacific region. Therefore, the BASE CBCs represent the background profile of the open ocean environment. In contrast, the remaining three scenarios utilize boundary conditions generated from the output of three global chemistry models (GCMs), namely, Hemisphere version of the Community Multiscale Air Quality model (H-CMAQ), Goddard Earth Observing System-Chemistry (GEOS-Chem), and Community Earth System Model version 2.2 (CESM2.2). Each of these boundary datasets was processed and formatted consistently to ensure compatibility with the CMAQ framework.

Specifically, CBCs for the H-CMAQ scenario were derived from daily averaged species



concentration outputs produced by a hemispheric CMAQ simulation under the U.S.
EPA's Air Quality Time Series (EQUATES) Project
(http://www.epa.gov/cmaq/EQUATES, last accessed: 1 August 2024). These
simulations were conducted using a customized version of CMAQ v5.3.2, with a
horizontal resolution of 108 × 108 km on a polar stereographic projection, and
employed the CB6R3M_AE7_KMTBR chemical mechanism.
For the GEOS-Chem scenario, 3-hourly global simulation outputs were used. The
GEOS-Chem model is a global 3-D chemical transport model driven by meteorological
fields from NASA's Goddard Earth Observing System (GEOS), developed by the
NASA Global Modeling and Assimilation Office. The chemical mechanism includes
comprehensive tropospheric $O_3$–$NO_x$–VOCs–aerosol–halogen chemistry (Mao et al.,
2013; Park et al., 2004; Parrella et al., 2012; Wang et al., 1998), as well as stratospheric
chemistry processes (Eastham et al., 2014). Further information is available at
https://geoschem.github.io/ (last accessed: 1 August 2024).
The CESM2.2 scenario utilized 6-hourly output from the Community Atmosphere
Model with Chemistry (CAM-chem) embedded within the Community Earth System
Model version 2.2 (CESM2.2). The CAM-chem simulations used the finite-volume
dynamical core, with a horizontal resolution of 1° × 1° and 32 vertical levels. The
MOZART-T1 mechanism was applied to simulate both tropospheric and stratospheric
chemical processes. Details on the model setup and outputs are available at
https://www2.acom.ucar.edu/gcm/cam-chem.
All the three global model outputs were converted to the I/O API format required by
the CMAQ Chemical Transport Model (CCTM). A combination of data transformation
tools and custom scripts was developed and applied to harmonize species mapping,
spatial resolution, temporal alignment, and file formatting, thus enabling seamless
integration of each global model dataset as boundary conditions for the regional CMAQ
simulations. To minimize the influence of initial conditions and allow the imposed
boundary conditions to fully propagate throughout the simulation domain, a 10-day
model spin-up period was applied prior to the analysis period.



**2.2 Observation data**

**2.2.1 Surface observation data**

Surface observations across China for July–August 2019 were used to evaluate the simulated meteorological parameters and atmospheric pollutant concentrations from the WRF-CMAQ model. Meteorological data were obtained from the National Meteorological Information Center (http://data.cma.cn, last accessed 1 January 2024). Hourly meteorological observations from 2,394 monitoring stations were collected, including 2-meter air temperature (T2), 2-meter relative humidity (RH2), 10-meter wind speed (WS10), and surface pressure (PRS). Hourly $O_3$ observations were retrieved from the China National Environmental Monitoring Center (https://air.cnemc.cn:18007/, last accessed 1 January 2024), encompassing data from 1,480 air quality monitoring sites. The spatial distribution of meteorological and air quality monitoring stations is shown in Fig. 1 and Fig. S1. To investigate the spatial variability of chemical boundary condition impacts on $O_3$ simulation, monitoring sites were grouped into seven regions within China (Fig. 1 and Table S1): South (S), East (E), North (N), Central (C), Northeast (NE), Northwest (NW), and Southwest (SW).

This study evaluates sites $O_3$ using the Maximum Daily 8-Hour Average concentration (O3MDA8), derived from both surface observations and model simulations. To comprehensively assess model performance across different pollution levels, we analyze two key indicators: the average O3MDA8 (avg-O3MDA8), which reflects the overall background and typical exposure level, and the 90th percentile of O3MDA8 (90th-O3MDA8), which is used to characterize high-$O_3$ events. The inclusion of the 90th percentile metric enables evaluation of the model's ability to capture peak $O_3$ pollution episodes that are most relevant to regulatory thresholds and public health risk assessments.

Model performance was quantitatively evaluated using multiple statistical metrics, including mean observed value (OBS), mean simulated value (SIM) for each of the four CBC scenarios (BASE, H-CMAQ, GEOS-Chem, CESM2.2), mean bias (MB), normalized mean bias (NMB), root mean square error (RMSE), index of agreement (IOA), and Pearson correlation coefficient (r). The mathematical definitions of these statistics are provided below.

$$\text{OBS} = \frac{1}{n}\sum_{i=1}^{n} O_i$$

$$\text{SIM} = \frac{1}{n}\sum_{i=1}^{n} S_i$$

$$\text{MB} = \frac{1}{n}\sum_{i=1}^{n} (S_i - O_i)$$

$$\text{NMB} = \sum_{i=1}^{n} (S_i - O_i) \Big/ \sum_{i=1}^{n} O_i$$



$$RMSE = \sqrt{\frac{1}{n}\sum_{i=1}^{n}(S_i - O_i)^2}$$

$$r = \frac{\sum_{i=1}^{n}(S_i - SIM)(O_i - OBS)}{\sqrt{\sum_{i=1}^{n}(S_i - SIM)^2}\sqrt{\sum_{i=1}^{n}(O_i - OBS)^2}}$$

$$IOA = 1 - \frac{n*RMSE^2}{\sum_{i=1}^{n}(|S_i - OBS| + |O_i - OBS|)^2}$$

where $S_i$ and $O_i$ are the simulated and observed site's avg-O3MDA8 or 90th-
O3MDA8, n represents the number of the simulated days.
**2.2.2 Vertical Observation Data**
To evaluate the influence of CBCs on the vertical distribution of $O_3$, $O_3$ sounding data
from five representative sites (Hong Kong, Nanjing, Golmud, Lhasa, and Lijiang) were
collected and used to validate the model's vertical $O_3$ simulations. These stations are
strategically located in the eastern, southern, southwestern, and northwestern
boundaries of the modeling domain (Fig 1), enabling a targeted assessment of how
boundary conditions affect $O_3$ concentrations aloft. Briefly, the Hong Kong profiles
(King's Park Observatory; launched at 13:00 LST; 9 soundings during July–August
2019) were obtained from the World Ozone and Ultraviolet Radiation Data Centre
(WOUDC, https://woudc.org/data.php, last accessed: October 8, 2024). Nanjing
observations (from the National Benchmark Climate Station; launched between 13:00
and 14:00 LST; 4 soundings between July 23, 2019 and September 1, 2020) were
sourced from the China Air Pollution Data Center (CAPDC,
https://www.capdatabase.cn, last accessed: October 8, 2024). Data for Golmud (12
profiles), Lhasa (8 profiles), and Lijiang (5 profiles), collected between 2019 and 2022
with launch times ranging from 23:00 to 02:00 LST, were obtained from the National
Tibetan Plateau Data Center (TPDC, https://data.tpdc.ac.cn/home, last accessed:
October 10, 2024) (Bai Zhixuan, 2023; Zhixuan, 2023; Bai Zhixuan, 2022). To ensure
consistency across datasets and comparability with the model output, all sonde data
were processed for the 0–20 km altitude range and interpolated to match the model
vertical structure. Observations during July–August were prioritized, and model outputs
were extracted as time-averaged vertical profiles over the corresponding grid cells and
times (13:00 -14:00 or 23:00 - 2:00 LST). Detailed information about the surface
observation and sounding sites is provided in Table S2.
In addition, tropospheric $O_3$ column data were also introduced to further evaluate the
spatial performance of the model. This dataset was developed by the University of
Science and Technology of China (USTC) and is derived from measurements by the
Environmental Trace Gases Monitoring Instrument (EMI) aboard the Gaofen-5 satellite,
China's first ultraviolet-visible hyperspectral spectrometer dedicated to atmospheric
composition monitoring. The product provides a seamless tropospheric $O_3$ column
dataset at a high spatial resolution of 1 km × 1 km, offering detailed information on $O_3$





distribution over complex terrains and urban regions. Further details on the retrieval
algorithm and validation of the product can be found in (Zhao et al., 2024). Detailed
information about the Tropospheric O₃ column data is also provided in Table S2.

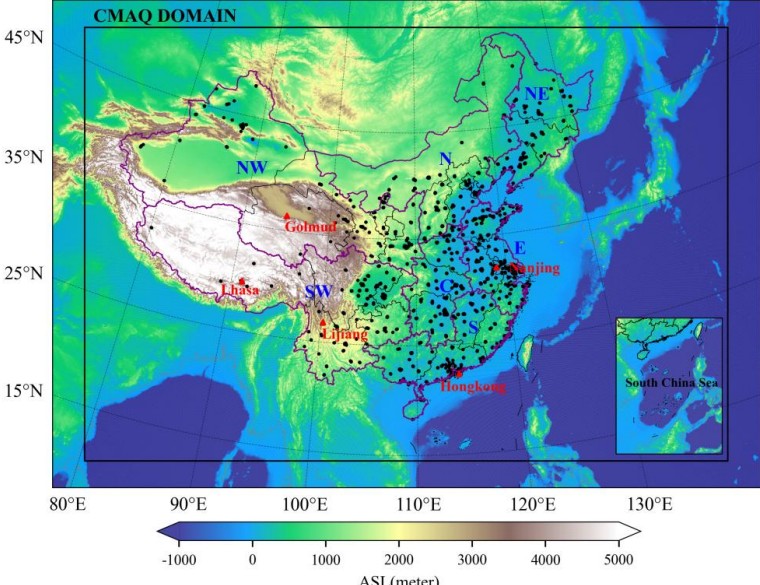


**Figure 1.** Simulation domain of the CMAQ model with a horizontal resolution of 36 × 36 km. Black dots
represent surface O₃ monitoring sites, and red triangles denote O₃ sounding launch stations. Terrain
elevation above sea level (ASL) is illustrated with shaded relief. Purple lines delineate the administrative
boundaries of China's major regions—South (S), East (E), North (N), Central (C), Northeast (NE),
Northwest (NW), and Southwest (SW). The provinces included in each region are listed in Table S1.
**3. Results and Discussions**
**3.1 Comparative Analysis of four CBCs**
Fig. 2 displays the vertically averaged temporal O₃ distribution along the four lateral
boundaries of the modelling domain during July-August 2019, under four different
CBC scenarios. In the BASE scenario, the O₃ profile remains static, characterized by
horizontally uniform mixing ratios at each altitude. A sharp increase in O₃ concentration
is evident near the tropopause, with minimal vertical variation in the lower and mid-
troposphere. This relatively uniform, three-dimensional O₃ distribution suggests that
the BASE CBC scenario represents a background condition (i.e, over the open ocean),
and thus fails to adequately capture the spatiotemporal variability of O₃ over mainland
East Asia, where O₃ levels are strongly influenced by anthropogenic emissions and
regional transport processes.
In contrast to the static pattern in the BASE scenario, the O₃ boundary conditions
extracted from the three global models (H-CMAQ, GEOS-Chem, and CESM2.2)
exhibit both horizontal and vertical variability across the four lateral boundaries. These



three scenarios display a generally consistent spatial and vertical structure. However, notable differences still exist across different boundaries. In the lower troposphere (0–3 km), the average $O_3$ concentrations from the three global models are 5–7 ppbv lower than those in the BASE scenario along the southern and eastern boundaries, with only minor differences among the models. For instance, over the oceanic portions of these boundaries, specifically the eastern segment of the southern boundary and the southern segment of the eastern boundary, the BASE scenario overestimates boundary $O_3$ concentrations by as much as 20–30 ppbv. In contrast, along the northern and western boundaries, the global models generally produce 4–20 ppbv higher $O_3$ concentrations than the BASE scenario, accompanied by greater inter-model variability. Among them, H-CMAQ and GEOS-Chem show relatively similar patterns, whereas CESM2.2 exhibits substantially higher $O_3$ levels, particularly along the western boundary (Fig. 2 and Table 1).

Conversely, compared to the BASE scenario, the differences in boundary $O_3$ concentrations among the three global models significantly increased in the mid-to-upper troposphere (3-10 km) and stratosphere (above 10 km). In the mid-to-upper troposphere (3-10 km), the BASE scenario generally underestimated $O_3$ concentrations along the northern and western boundaries, while significantly overestimating them along the southern boundary. The CESM2.2 scenario showed higher $O_3$ concentrations along the eastern, northern, and western boundaries. In the stratosphere (above 10 km), the global model results indicated that the BASE scenario significantly overestimated $O_3$ concentrations along the southern, eastern, and western boundaries, with GEOS-Chem exhibiting the lowest $O_3$ concentrations among the four scenarios. The H-CMAQ and CESM2.2 models showed large areas of high $O_3$ concentration near the northern boundary. These spatial variations in $O_3$ boundary conditions are likely to have a considerable impact on the simulation of surface $O_3$ concentrations over China during the summer months.

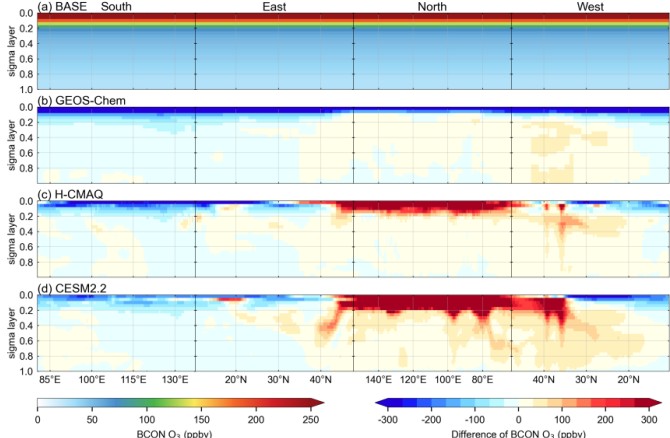

**Figure 2.** Temporally averaged $O_3$ chemical boundary conditions along the lateral boundaries of CMAQ modeling domain. Panels show (a) BASE scenario $O_3$ concentrations, (b) GEOS-CHEM minus BASE , (c) H-CMAQ minus BASE, and (d), CESM2.2 minus BASE. Values are plotted clockwise, starting from the southwest corner of the CMAQ simulation domain, with the model's sigma coordinates.



**Table 1.** Statistical results of average O$_3$ concentrations(ppbv)at various vertical heights among the four boundaries for four CBC scenarios.

| Vertical altitude | Boundary | BASE | H-CMAQ | GEOS-Chem | CESM2.2 |
|---|---|---|---|---|---|
| 0-3 km | South | 31.7 | 27.3 | 25 | 26 |
|  | East | 31.7 | 24.9 | 23.8 | 25.1 |
|  | North | 31.7 | 39.5 | 36.4 | 47.4 |
|  | West | 31.7 | 43.5 | 45.8 | 51.7 |
| 3-10 km | South | 53.2 | 40.1 | 26.7 | 35.6 |
|  | East | 53.2 | 54.9 | 43.4 | 61.7 |
|  | North | 53.2 | 79.2 | 68.4 | 119.7 |
|  | West | 53.2 | 76.2 | 60.6 | 88.7 |
| Above 10 km | South | 408.0 | 233.4 | 42.4 | 272.0 |
|  | East | 408.0 | 351.6 | 82.0 | 373.8 |
|  | North | 408.0 | 658.3 | 186.9 | 728.7 |
|  | West | 408.0 | 338.6 | 88.2 | 324.1 |

## 3.2 Evaluation of Model Performance Using Different CBCs

### 3.2.1 Meteorological simulation evaluation

Table 2 presents an evaluation of WRF model simulations of 2-meter temperature (T2), 2-meter daily maximum temperature (T2max), 2-meter relative humidity (RH2), 10-meter wind speed (WS10), surface pressure (PRS), and precipitation (PRECIP). The data were averages from 2439 meteorological stations across China. Analysis of mean bias (MB), correlation coefficient (r), and index of agreement (IOA) revealed that the WRF model accurately simulated the meteorological fields. T2, T2max, RH2, and PRS exhibited IOA values exceeding 0.85, indicating strong agreement with observations. Correlation coefficients (r) exceeded 0.7 for all variables except WS10. However, some biases remained in the simulation results for certain variables. Specifically, RH2 and PRS were slightly underestimated, while PRECIP was overestimated; nevertheless, the r and IOA values remained relatively high. In contrast, WS10 was significantly overestimated (by 1.6 m/s), with both IOA and r below 0.5. This likely stemmed from the relatively coarse model grid resolution, hindering accurate representation of urban topography and its impact on wind speed—a phenomenon observed in other studies (Hu et al., 2016; Shen et al., 2022). Overall, the WRF model demonstrated good performance in meteorological simulations, providing reliable inputs for the CMAQ model.






**Table 2.** Evaluation results for the meteorological variables.

| variable | OBS | SIM | MB | RMSE | IOA | r |
|---|---|---|---|---|---|---|
| T2 (℃) | 25 | 24.4 | -0.6 | 2.1 | 0.94 | 0.91 |
| T2max (℃) | 29.9 | 29.3 | -0.6 | 2.5 | 0.91 | 0.86 |
| RH2 (%) | 73.6 | 69.1 | -4.5 | 8.6 | 0.88 | 0.86 |
| WS10 (m/s) | 2 | 3.6 | 1.6 | 1.8 | 0.41 | 0.45 |
| PRS (hPa) | 937.8 | 922.7 | -15.1 | 28.5 | 0.97 | 0.97 |
| PRECIP (mm) | 297.4 | 434.3 | 136.9 | 234.2 | 0.72 | 0.7 |

**3.2.2 Surface O₃ Simulation Performance**
Fig. 3 illustrated the spatial distribution of avg-O3MDA8 and 90th-O3MDA8
concentrations and their normalized mean bias (NMB) across China. Across all
monitoring sites, the observed avg-O3MDA8 and 90th-O3MDA8 for July-August 2019
were 59.4 ppbv and 82.8 ppbv, respectively (Table S3). Generally, $O_3$ concentrations in
North China were higher than in South China. For 90th-O3MDA8, elevated values were
widespread, notably in the North China Plain (NCP), Central China, Yangtze River
Delta (YRD), Pearl River Delta (PRD), and Sichuan Basin (SCB), highlighting the
severity of summer $O_3$ pollution across China.
Substantial discrepancies existed between observed and simulated $O_3$ across all CBC
scenarios. The BASE scenario, in particular, systematically underestimated both mean
and extreme $O_3$, especially in northern regions (latitude > 30° N). Avg-O3MDA8
underestimations reached –16.1 ppbv in North China and –11.2 ppbv in Northwest
China, with moderate underestimations in East (–5.1 ppbv) and Northeast China (–4.8
ppbv) (Fig. 3; Table S4). Similarly, 90th-O3MDA8 was underestimated by 32.9% in
North China and 26.2% in Northwest China, with smaller NMB  (-9.3 to -14.2%)
elsewhere except South China (Table S5). These results indicate that the BASE scenario
poorly represents both average and high $O_3$ levels, limiting its ability to capture $O_3$
formation and transport processes during hot seasons.
By incorporating global model-derived CBCs, significant improvements in both bias
and agreement are observed across China. Based on the NMB values for avg-O3MDA8
and 90th-O3MDA8, the three dynamic CBC scenarios can be ranked as follows: GEOS-
Chem (–0.3%, –6.5%) > H-CMAQ (–1.1%, –7.9%) > CESM2.2 (+4.9%, –0.7%)
(Tables S4-S5). GEOS-Chem consistently yielded the smallest bias, indicating the most
accurate representation of boundary and background $O_3$ inflow at both average and
extreme levels. Correspondingly, its index of agreement (IOA) reached 0.85 for avg-
O3MDA8 and 0.83 for 90th-O3MDA8, the highest among all scenarios, suggesting
excellent spatial and temporal consistency with observations. The H-CMAQ scenario
also improved upon the BASE case, albeit to a slightly lesser extent, reducing O3
underestimation while maintaining IOAs of 0.82 (avg-O3MDA8) and 0.81 (90th-
O3MDA8). In contrast, the CESM2.2 scenario exhibited a positive NMB for avg-
O3MDA8 (+4.9%), suggesting a slight overestimation in background inflow. However,
CESM2.2 substantially improved the simulation of $O_3$ extremes, with a much smaller
NMB (–0.7%) for 90th-O3MDA8 and a still-high IOA of 0.83, highlighting its strength
in reproducing high-$O_3$ pollution events, especially in regions influenced by complex
terrain and strong photochemistry. Overall, these results demonstrate that applying



dynamic CBCs derived from global chemical transport models substantially enhances
the simulation of both average and extreme O₃ concentrations.

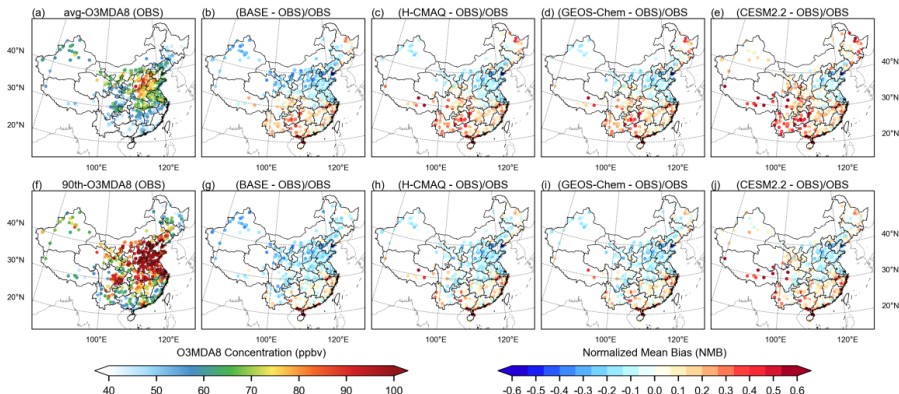


**Figure 3.** Spatial distribution of avg-O3MDA8 and 90th-O3MDA8 from observations (OBS) and four
CBC scenario simulations. Panels (a, f) show observed avg-O3MDA8 and 90th-O3MDA8 at 1,480
monitoring sites, while panels (b–e, g–j) present site-level normalized mean bias (NMB) for BASE, H-
CMAQ, GEOS-Chem, and CESM2.2 simulations, respectively.

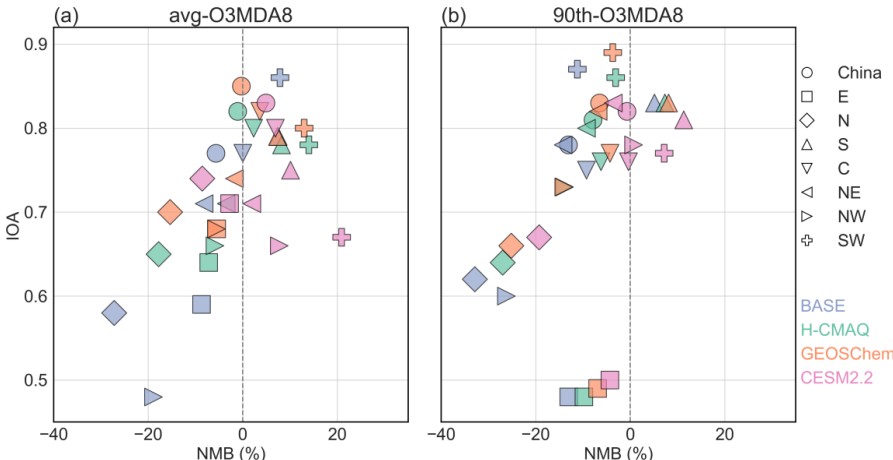

458

**Figure 4.** Comparison of model performance among four CBC scenarios (BASE, H-CMAQ, GEOS-
Chem, and CESM2.2) in terms of Normalized Mean Bias (NMB, %) and Index of Agreement (IOA) for
(a) average daily maximum 8-hour O₃ concentrations (avg-O3MDA8) and (b) the 90th percentile of daily
maximum 8-hour O₃ (90th-O3MDA8) in China and its seven subregions (South (S), East (E), North (N),
Central (C), Northeast (NE), Northwest (NW), and Southwest (SW)).

At the regional scale, however, differences among the three dynamic CBC scenarios
become regionally differentiated (Fig. 4). Although GEOS-Chem and H-CMAQ
consistently show the best nationwide performance, CESM2.2 demonstrates superior
accuracy in several regions. For instance, CESM2.2 achieves the smallest NMB and
highest IOA in the north (N), northeast (NE), east (E) and northwest (NW) regions for
both avg-O3MDA8 and 90th-O3MDA8, reflecting its strength in capturing high O₃



events in areas. In the SW region, CESM2.2 outperforms other models with a positive
NMB yet high IOA, indicating a well-aligned simulation of elevated background $O_3$
levels. In contrast, GEOS-Chem exhibits balanced performance across most regions,
notably achieving relatively low NMB and high IOA in the east (E), central (C), and
northeastern (NE) regions. These areas are typically influenced by continental outflow
and moderate photochemistry, conditions under which GEOS-Chem's background $O_3$
input appears to be well-optimized. H-CMAQ offers moderate improvement relative to
the BASE scenario across most regions, with less extreme biases than BASE and
slightly lower IOA compared to CESM2.2 or GEOS-Chem.
**3.2.3 Vertical $O_3$ profile Evaluation**

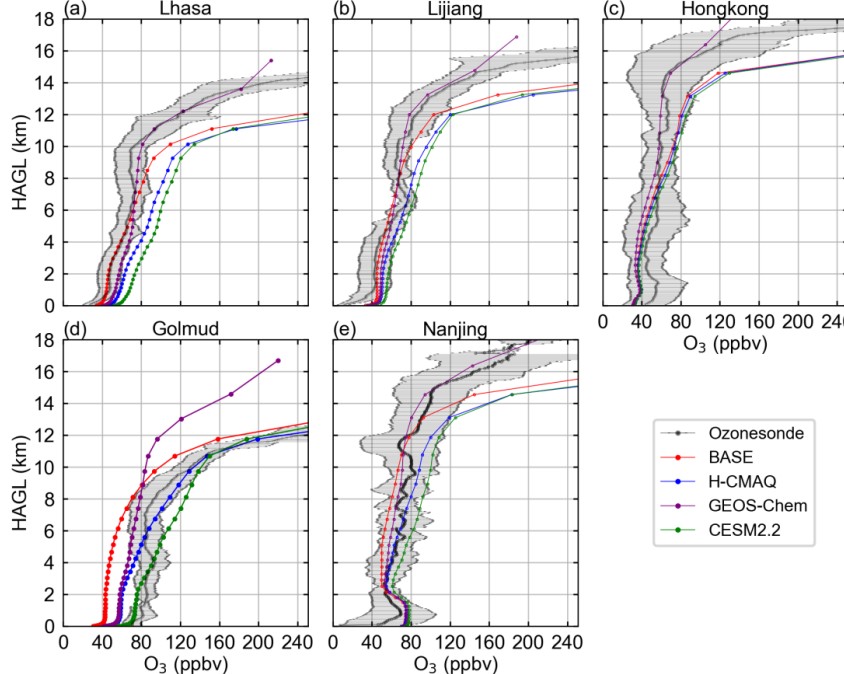


**Figure 5.** Comparison of vertical $O_3$ profiles between four CBC scenario simulation (BASE, H-CMAQ,
GEOS-Chem, and CESM2.2) and sounding observations at five stations across China.




**Table 3** Comparison and evaluation of vertical $O_3$ concentration profiles (ppbv) at each sounding station.

|  | Lhasa | Lijiang | Hongkong | Golmud | Nangjing |
|---|---|---|---|---|---|
| Lower troposphere (0-3 km) | | | | | |
| OBS | 45.4 | 39.1 | 49.9 | 64.9 | 57.9 |
| BASE | 45.3 | 45.1 | 40.8 | 43.0 | 64.8 |
| H-CMAQ | 60.6 | 50.6 | 41.3 | 59.4 | 67.0 |
| GEOS-Chem | 54.8 | 48.8 | 39.5 | 57.5 | 67.3 |
| CESM2.2 | 70.4 | 55.7 | 41.9 | 74.6 | 71.0 |
| Mid-to-upper troposphere (3-10 km) | | | | | |
| OBS | 66.7 | 62.1 | 53.9 | 85.0 | 68.2 |
| BASE | 75.1 | 62.0 | 51.4 | 62.7 | 56.1 |
| H-CMAQ | 94.9 | 74.8 | 53.9 | 94.7 | 71.5 |
| GEOS-Chem | 72.6 | 62.9 | 45.8 | 73.8 | 62.2 |
| CESM2.2 | 104.0 | 81.4 | 56.7 | 112.0 | 83.5 |
| lower stratosphere (10-16 km) | | | | | |
| OBS | 230.8 | 122.1 | 68.1 | 174.6 | 92.3 |
| BASE | 306.6 | 211.7 | 106.5 | 288.1 | 117.7 |
| H-CMAQ | 394.4 | 259.3 | 112.8 | 367.7 | 149.7 |
| GEOS-Chem | 147.5 | 106.6 | 66.8 | 131.8 | 85.7 |
| CESM2.2 | 371.3 | 249.3 | 116.0 | 332.6 | 153.8 |

To assess the performance of the model in simulating vertical $O_3$ distribution under different Chemical Boundary Conditions (CBCs), we compared the simulated $O_3$ profiles from the four scenarios with observational data from five ozonesonde stations (Fig. 5). To better evaluate model performance at different altitudes, we computed mean $O_3$ concentrations within three representative vertical layers: the lower troposphere (0–3 km), the middle-to-upper troposphere (3–10 km), and the lower stratosphere (10–16 km), as summarized in Table 3.

In lower troposphere (0–3km), observed $O_3$ concentrations in this layer ranged from ~39 to 65 ppbv across the five sites. $O_3$ concentrations in the lower troposphere are sensitive to both local emissions and background inflow. The BASE scenario generally underestimated $O_3$, whereas incorporating dynamic CBCs increased near-surface concentrations. Among the scenarios, GEOS-Chem exhibited the most balanced performance, with mean biases typically within ± 10 ppbv. CESM2.2 overestimated $O_3$ substantially at high-altitude sites, e.g., +25.0 ppbv at Lhasa and +9.7 ppbv at Golmud, indicating excessive inflow of near-surface $O_3$. H-CMAQ also slightly overestimated $O_3$, but with smaller magnitudes. These results indicate that while dynamic CBCs improve near-surface $O_3$ representation, overestimation in clean or elevated regions (e.g., Lhasa) must be carefully considered, especially when using CESM2.2.

The mid-to-upper troposphere (3~10km) reflects regional background transport, deep convection, and vertical mixing. Observed $O_3$ levels typically increased with altitude, ranging from ~54 to 85 ppbv. The BASE scenario consistently underestimated $O_3$ in this layer, particularly in Golmud (–22.3 ppbv) and Nanjing (–12.1 ppbv), due to



insufficient $O_3$ inflow. Dynamic CBCs significantly reduced this bias. H-CMAQ and CESM2.2 both improved model–observation agreement, but CESM2.2 often overestimated $O_3$ (e.g., +37.3 ppbv in Lhasa), potentially reflecting overly strong entrainment of free-tropospheric $O_3$. GEOS-Chem again performed best overall, producing values close to observations in Lijiang, Nanjing, and Lhasa, demonstrating a good balance between under- and overestimation. This suggests that GEOS-Chem CBCs offer the most realistic representation of free-tropospheric $O_3$, while CESM2.2 may be too aggressive in polluted or convective regions.

In the lower stratosphere (10–16 km), $O_3$ levels increased sharply in this layer, with observed values ranging from ~68 to 231 ppbv. The BASE scenario significantly overestimated stratospheric $O_3$ at all sites, especially in Golmud and Lhasa, indicating excessive intrusion of stratospheric $O_3$ in default boundary inputs. This bias was further amplified in H-CMAQ and CESM2.2, with overestimations exceeding ~164 ppbv and ~140 ppbv in Lhasa and ~193 ppbv and ~158 ppbv in Golmud. From the vertical $O_3$ profile comparison, the elevated biases in the lower stratosphere of BASE, H-CMAQ, and CESM2.2 scenarios may enhance the stratosphere–troposphere exchange (STE), especially in southwestern and southern China, which is significantly influenced by the Qinghai-Tibet Plateau (Fig. 5a-c,e). In contrast, GEOS-Chem was the only CBC that consistently reduced this overestimation, bringing modeled values closer to observations at all sites. For example, it lowered the stratospheric bias in Golmud from +113.5 ppbv (BASE) to –42.8 ppbv and achieved near-perfect agreement in Hong Kong (-1.3 ppbv). Overall, the vertical profile analysis underscore that GEOS-Chem provides the most accurate representation of upper tropospheric and stratospheric $O_3$ inflow, especially important for western China where STE processes are more active.

### 3.2.4 Satellite-Based O3 Column Assessment

The spatial distribution of tropospheric ozone column (TOC) concentrations provides valuable insights into regional $O_3$ pollution patterns. In this study, TOC concentrations retrieved from the Environmental Trace Gases Monitoring Instrument (EMI) aboard the Gaofen-5 satellite during July–August 2019 were compared with simulation (SIM) results from four different scenario models (Fig. 6). Observational data from EMI indicate a general increase in TOC concentrations with latitude across China, consistent with previous studies (Zhu et al., 2022; Liu et al., 2022). North China exhibits the highest TOC values among the eastern regions, corresponding to areas known for severe surface-level $O_3$ pollution (Lu et al., 2018).

From the numerical modeling perspective, the simulation scenarios based on the BASE, H-CMAQ, and CESM2.2 models predominantly reflect an overestimation of TOC concentrations. Among them, the BASE scenario demonstrates the least degree of overestimation, particularly in the South China and Northeast China regions, where overestimations range from 20 to 30 DU, while other regions exhibit overestimations between 10 and 20 DU. Both the H-CMAQ and CESM2.2 models show robust overestimations exceeding 20 DU, especially in northern China (north of 35°N), where the overestimation can surpass 40 DU, with the Northeast region registering the highest overestimation, reaching beyond ~50—60 DU. In contrast, the CBCs boundary conditions provided by the GEOS-Chem model yield superior results in simulating the



spatial distribution of TOC, with slight overestimations noted in South China and areas
north of 40°N, while underestimating concentrations within the latitude range of 30°N
to 40°N. The regional mean bias (MB) of model-simulated TOC versus satellite
observations was calculated for the mainland of China. The MB of model-simulated
TOC(DU)for the four scenarios—BASE, H-CMAQ, CESM2.2, and GEOS-Chem—
were 14.4 DU, 40.4 DU, 41.7 DU, and 0.7 DU, respectively, consistent with the analysis
results shown in Fig. 5. And the simulation discrepancies for TOC across China are
confined to approximately ±10 DU, indicating that GEOS-Chem's CBCs represent the
optimal boundary condition input for regional O$_3$ modeling in China area.

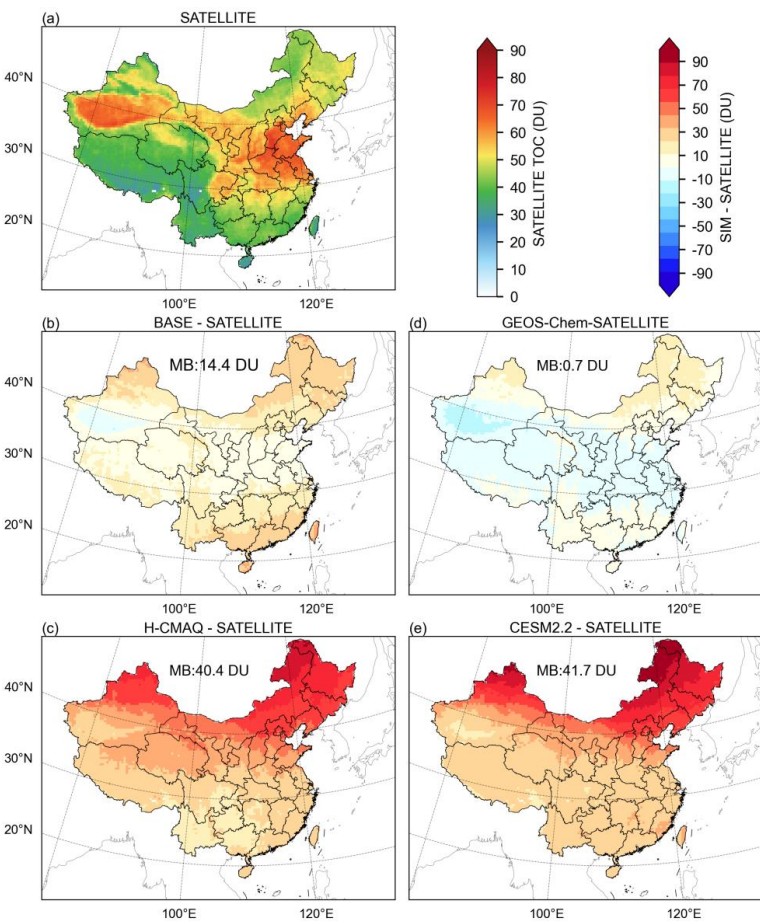


**Figure 6.** Comparison of tropospheric ozone column (TOC) distributions over China between satellite
observations and model simulations. Panel (a) shows the TOC retrieved from satellite measurements,
while panels (b–e) depict the differences (MB) between simulated TOC from the BASE, H-CMAQ,
GEOS-Chem, and CESM2.2 scenarios and the satellite retrieval.



**3.3 Mechanism of the impact of CBCs on O₃ formation**

**3.3.1 General impact of synoptic-scale circulation**

CBCs regulate regional O₃ by controlling the inflow of background O₃ and precursors at model boundaries. Given the superior performance of GEOS-Chem in reproducing surface and vertical O₃ based on our validations, we further contrast GEOS-Chem with the BASE scenario to highlight the role of CBCs in cross-boundary transport at the surface and at 700, 500, and 200 hPa isobaric surfaces (Fig. 7).

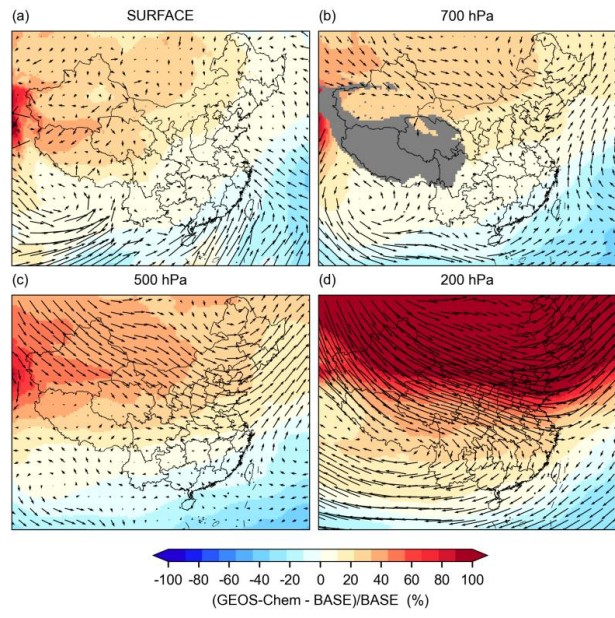

**Figure 7.** Normalized mean bias (NMB, (GEOS-Chem − BASE)/BASE) of mean O₃ concentrations and corresponding mean flow fields at surface and 700, 500, and 200 hPa isobaric surfaces over the simulation domain. (The grey area indicates invalid value.)

In southeastern China, summer monsoonal flow carried relatively clean marine air into the mid–lower troposphere, lowering background O₃ and suppressing accumulation over eastern and southern regions, where GEOS-Chem boundary conditions produced slightly reduced concentrations (<4%, Fig. 7a-7b). This dilution effect is consistent with the characteristic influence of the Western Pacific Subtropical High during summer, which effectively flushes the coastal boundary layer with cleaner oceanic air masses. In contrast, along the northern and western boundaries, GEOS-Chem introduced substantially higher O₃ than BASE. Advected by prevailing northwesterlies, these inflows penetrated deep into inland China, increasing surface O₃ by more than 10% across most regions and by over 20% in northern and northwestern China. The magnitude of this enhancement aligns with the recognized impact of long-range transport from Eurasia, which often elevates the ozone baseline in northern China. The influence of CBC was even stronger at higher altitudes (Fig. 7c-7d). At 500 and 200 hPa, GEOS-Chem introduced markedly higher O₃ than BASE, reflecting enhanced



background inflows and contributions from stratospheric air masses. This vertical gradient in CBC sensitivity underscores the role of the free troposphere as a reservoir for long-lived $O_3$. Such upper-level enhancements have important surface implications, as downward mixing and stratosphere–troposphere exchange (STE) can transport high-$O_3$ air into the boundary layer under favorable meteorological conditions, especially during the passage of cold fronts or deep convective mixing, further exacerbating pollution episodes.

Overall, these results highlight that the mechanistic impact of CBC on $O_3$ formation arises from a synergistic combination of boundary inflow composition and large-scale circulation. While oceanic inflows tend to dilute $O_3$ in southern and eastern regions, strong continental and stratospheric inflows from the north and west can significantly elevate both free-tropospheric and surface $O_3$, amplifying pollution severity in inland China. These findings confirm that accurate CBCs are not merely a model constraint but a vital component for capturing the dynamic interplay between local photochemistry and global atmospheric circulation.

### 3.3.2 Case Study: Synoptic-Scale Circulation Dynamics Modulating CBC Impacts

The influence of CBC varies dynamically with large-scale meteorological conditions rather than remaining static. During summer, synoptic disturbances such as the Western Pacific Subtropical High extensions, tropical cyclone activity, and East Asian westerly jet fluctuations reshape regional circulation patterns and modulate the transport of polluted or clean air masses into the model domain. These circulation changes, characterized by alternating cyclonic and anticyclonic flows, substantially alter the efficiency of transboundary transport and consequently affect CBC impacts on near-surface $O_3$ simulations. Here, we examined two sequential circulation regimes during August 2019 associated with successive typhoon events: Super Typhoon Lekima and Typhoon Krosa. These events created distinctly different transport patterns that modulated how boundary conditions influenced surface $O_3$ across China. Based on this evolution, we define two phases: Phase 1 (P1, 10–14 August, during Lekima's landfall and decay in Eastern China) and Phase 2 (P2, 15–19 August, controlled by post-trough northwesterlies) (Fig. 8).



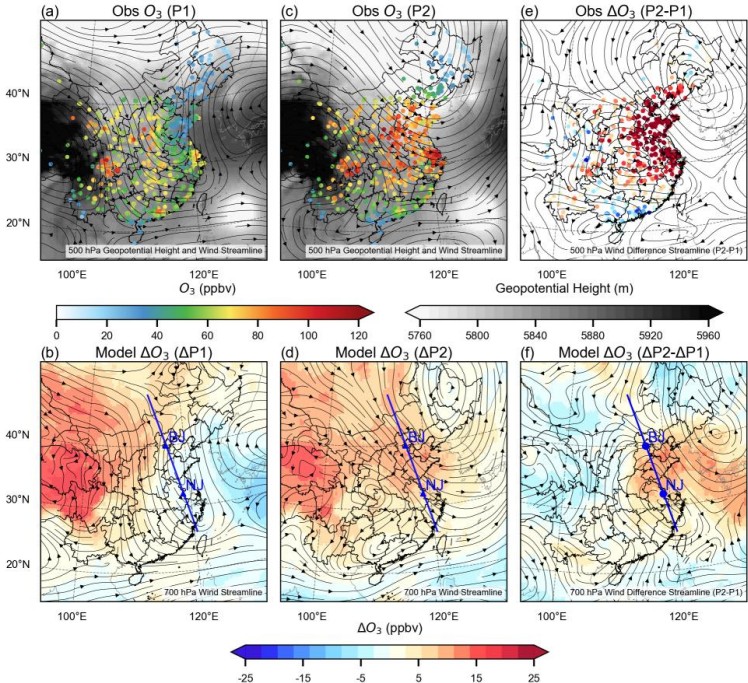

**Figure 8.** (a) Distribution of 500 hPa winds (streamlines), geopotential height (contours), and surface O₃ observations (dots) during P1; (b) Distribution of 700 hPa winds (streamlines), and difference in modeled surface O₃ between GEOS-Chem CBC and BASE during P1; (c) same as (a) but for P2; (d) same as (b) but for P2; (e) differences in observed surface O₃ and 500 hPa winds between P1 and P2; (f) differences in simulated surface O₃ and 700 hPa winds between P1 and P2. Blue lines indicate the locations of vertical cross-sectional analyses, extending from north to south through Beijing (BJ) and Nanjing (NJ).

P1 occurred during the landfall and decay of Super Typhoon Lekima over the Yangtze River Delta (Fig 8a and 8b, and Fig. S2). The tropospheric circulation was dominated by a deep trough linked to the typhoon's remnant system, which enhanced southeasterly flow of marine air into eastern China. This pattern promoted deep convection and vigorous vertical mixing, leading to a pronounced coastal-inland gradient in surface O₃: marine-influenced coastal areas exhibited low concentrations (<50 ppbv), while inland regions maintained moderate-to-high levels (60–90 ppbv). The cyclonic circulation disrupted typical westerly transport pathways, reducing transboundary O₃ influence from northern and western source regions. Consequently, the BASE scenario overestimated O₃ by 15-25 ppbv over oceanic regions where static boundary conditions failed to capture typhoon-enhanced marine influence, while underestimating concentrations by 10-20 ppbv in northwestern China where continental transport remained active but was inadequately represented by the Pacific-based boundary profile (Fig. 8b).

By contrast, P2 was characterized by a dominant northwesterly flow across central and eastern China, situated behind a mid-level trough, while a high-pressure system strengthened over western China (Fig. 8c-8d). This "east-trough, west-ridge" configuration facilitated the efficient advection of O₃-rich air from western and northern




source regions, resulting in the noticeable $O_3$ elevations observed across the region (Fig. 8c-d). Model sensitivity analysis confirms that accurately representing these high-$O_3$ boundary inflows under such transport-favorable conditions elevates surface concentrations by 10–15 ppbv in the most affected areas (Fig. 8d). These results demonstrate that the BASE scenario, employing static boundary conditions, systematically underestimates cross-boundary pollution contributions during dynamically active periods when long-range transport is of importance.

The difference between P2 and P1 (Fig. 8e–f) illustrates a marked meteorological transition from a pollution-scavenging cyclonic regime during P1 to a pollution-accumulating regime in P2, characterized by trough-driven northwesterly transport and high-pressure-induced stability. This synoptic shift corresponded with observed surface $O_3$ increases of 30–60 ppbv across northern and central-eastern China. These regions aligned spatially with the anticyclonic circulation, where enhanced subsidence favored the accumulation of transported $O_3$. By incorporating chemically realistic CBCs, the simulation attributes approximately 10 ppbv of this $O_3$ increase to cross-boundary transport during P2 (Fig. 8f), highlighting the essential role of CBCs in accurately capturing $O_3$ buildup under transport-favorable synoptic regimes.

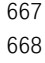
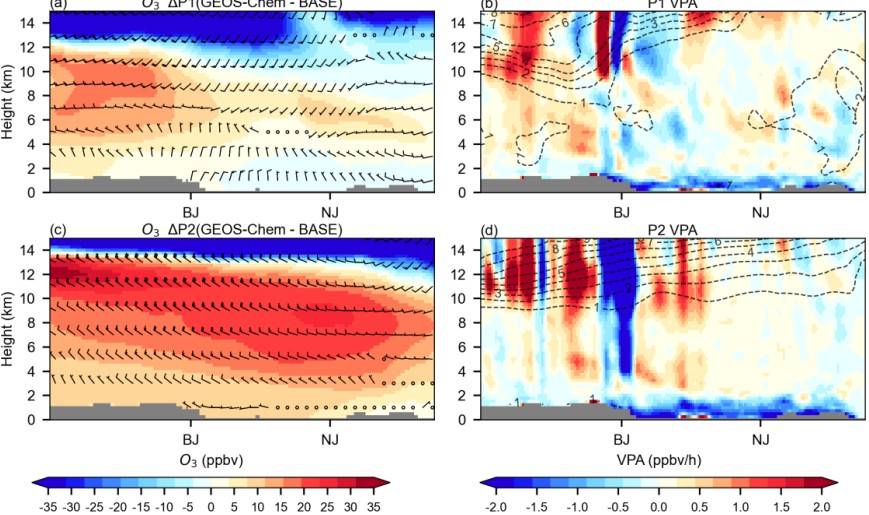

**Figure 9.** (a) Vertical cross-sectional analysis of the $O_3$ difference between GEOS-Chem CBC and the BASE scenario during P1, the wind bar denotes vertical wind field; (b) vertical distribution of potential vorticity (PV, dashed contours) and vertical transport (VPA, calculated by CMAQ process analysis as the sum of vertical diffusion and vertical advection) during P1; (c) same as (a) but for P2. (d) same as (b) but for P2. The x-axis labels BJ and NJ indicate the locations of Beijing and Nanjing, respectively.

To further clarify the role of dynamic CBC in $O_3$ simulations, we performed vertical cross-sectional analyses using the CMAQ process analysis module along the major



transport pathways during P1 and P2 (Fig. 9). Both phases consistently revealed strong
cross-boundary transport, with upstream inflows from outside the domain substantially
influencing downstream $O_3$ levels across mainland China. During P1, Typhoon Lekima
disrupted the transport corridor near the Yangtze River Delta (approximately 0–4 km),
restricting cross-boundary influences mainly to northern inflows affecting the North
China Plain (Fig. 9a). In contrast, under post-trough northwesterly flow during P2,
cross-boundary transport extended southward from the northern boundary, reaching as
far as the Yangtze River Delta (Fig. 9b). The difference between P1 and P2 highlights
a distinct transport corridor extending from higher to lower latitudes and from the mid–
upper troposphere toward the surface (Fig. S3), further emphasizing the crucial role of
dynamic CBCs in shaping $O_3$ distributions.
Here, we demonstrate that cross-boundary transport also occurs in the vertical
dimension, with $O_3$-rich air descending from the upper to the lower troposphere, while
stratosphere–troposphere exchange (STE) provided an additional pathway for
transboundary inflow. To identify possible STE occurrences, potential vorticity (PV)
between 10 and 14 km (above sea level) was examined, adopting a threshold of 2 PVU
(PV units, 1 PVU = $10^{-6}$ $m^{-2}$ $s^{-1}$ $kg^{-2}$) to distinguish stratospheric from tropospheric air
masses. STE events were evident over northern China during both P1 and P2 (Fig. 9c-
d and Fig. S4). In addition, the CMAQ process analyses with GEOS-Chem CBCs
corroborated intensified vertical transport between 10 and 14 km in both phases, with
distinctly positive contributions from vertical advection and turbulent diffusion. As a
result, the joint impact of large-scale advection and vertical mixing processes enabled
high-altitude $O_3$ to intrude into the lower troposphere and ultimately affect downstream
regions, even in YRD (such as Nanjing city).

## 4. Conclusion

This research demonstrates that CBCs represent a critical but often underappreciated
component of regional air quality modeling systems. We systematically evaluated the
influence of CBCs on regional $O_3$ simulations over China using the WRF-CMAQ
model. Four CBC scenarios were compared: a static BASE scenario using
climatological profiles and three dynamic scenarios derived from global chemical
transport models (H-CMAQ, GEOS-Chem, and CESM2.2). Overall, dynamic CBCs
substantially improved the representation of surface $O_3$ compared to the static BASE
scenario, with GEOS-Chem CBCs performing best. Across China, the normalized mean
bias (NMB) for avg-O3MDA8 was reduced from -5.7% (BASE) to -0.3% (GEOS-
Chem), and the index of agreement (IOA) increased from 0.77 to 0.85, while the 90th-
O3MDA8 percentile NMB improved from -13.1% to -6.5%, and the IOA increased
from 0.66 to 0.77. Based on ozonesonde profiles and satellite TOC evaluations, elevated
biases were identified in the lower stratosphere for BASE, H-CMAQ, and CESM2.2,
which may lead to overestimation of background $O_3$ concentrations, particularly during
STE events.
The influence of CBCs varies dynamically with large-scale meteorological conditions
rather than remaining static. During summer, synoptic disturbances such as the Western



Pacific Subtropical High extensions, tropical cyclone activity, and East Asian westerly jet fluctuations reshape regional circulation patterns and modulate the transport of polluted or clean air masses into the model domain. These circulation changes, characterized by alternating cyclonic and anticyclonic flows, substantially alter the efficiency of transboundary transport and consequently affect CBC impacts on near-surface $O_3$ simulations. Generally, oceanic inflows from the south dilute $O_3$ in southeastern and coastal areas, whereas strong continental and stratospheric inflows from northern and western boundaries significantly modulate tropospheric $O_3$, especially in downwind regions of the synoptic systems.

A comparative analysis of two successive synoptic regimes in July - August 2019, which shifted from a cyclone-dominated, pollution-scavenging phase to a post-trough northwesterly flow favorable for accumulation, revealed that dynamic circulation patterns enhanced cross-boundary transport both horizontally (via continental inflows from northern and western boundaries) and vertically (via stratosphere–troposphere exchange). The combined effects of these transport processes increased $O_3$ concentrations by 10–20% during high-pollution events over eastern China. These results underscore that accurate representation of dynamic CBCs is essential to capture circulation-driven horizontal and vertical transport and their integrated impact on regional $O_3$ distributions.

Our findings demonstrate that the choice of CBCs is not merely a technicality but a dynamic determinant of simulated $O_3$ levels for regional CTM, especially when facing synoptic regimes that favor long-range transport or vertical exchange. This underscores the necessity of moving beyond static boundary conditions in regional air quality modeling. To advance predictive capability, future efforts should pursue multi-model ensembles to quantify CBC uncertainty and explore the integration of real-time global fields into regional CTM forecasting systems. By elucidating the critical interplay between large-scale transport and regional pollution, this study provides a scientific foundation not only for improving $O_3$ forecasting but also for designing effective transboundary air quality management strategies.

## Acknowledgments

This research is supported by the National Key Research and Development Program (grant no. 2023YFC3709301), the National Natural Science Foundation Project (grant no. 42575120 and no. 42293322), the Youth Fund Project of the Sichuan Provincial Natural Science Foundation (24NSFSC2988), the Fundamental Research Funds for the Central Universities (Grant No. YJ202313). We acknowledge use of the hyperspectral remote sensing products of atmospheric compositions developed by Prof. Cheng Liu's group at the University of Science and Technology of China. The authors also thank the Tsinghua University for developing and sharing the MEIC emission inventory.

## Financial Support

This research is supported by the National Key Research and Development Program (grant no. 2023YFC3709301), the National Natural Science Foundation Project (grant no. 42575120 and no. 42293322), the Youth Fund Project of the Sichuan Provincial Natural Science Foundation (24NSFSC2988), the Fundamental Research Funds for the Central Universities (Grant No. YJ202313).



## Author Contributions

N.W. and F.Y. designed the research. Y.D. conducted the simulation. Y.D. and N.W. wrote the manuscript. S.L., Y.H., B.L. and G.S. contributed to the interpretation of the results. R.H., B.L., Y.J., N.W. and Y.F. provided critical feedback and helped to improve the manuscript.

## Competing Interests

The authors declare that they have no known competing financial interests or personal relationships that could have appeared to influence the work.

## Data Availability

The numerical simulation results were stored on Shuguang supercomputer, and results can be acquired from Nan Wang (nan.wang@scu.edu.cn)

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
