# Peer review of "The Role of Chemical Boundary Conditions in Simulating Summer Ozone and"

_EGUsphere, 2025_

## Referee Comment (RC1)

This study systematically evaluates the impact of different chemical boundary conditions (CBCs) on regional ozone simulations over China using the WRF-CMAQ model. This study results were validated against surface observation networks, ozonesonde profiles, and satellite data. The findings confirm the significant role of synoptic-scale circulation-driven transboundary transport in regional ozone simulation, air quality forecasting, and pollution management in China. The manuscript presents a well-structured and compellingly argued study, where the conclusions are fully substantiated by the analytical work. Some issues should be addressed before the consideration of publication.

1. In the manuscript, the term *chemical boundary conditions (CBCs)* suffers from redundant redefinition or inconsistent use of its abbreviation after initial definition. For example, *chemical boundary condition (CBCs)* is first defined on line 34; therefore, the abbreviation "CBCs" should be used directly in subsequent mentions (e.g., lines 127 – 128, 148, and 256). Conversely, the redefinition of *chemical boundary condition (CBCs)* on line 457 is unnecessary and should be removed. Moreover, both "chemical boundary condition" and "CBC" appear multiple times in the manuscript, and their usage is inconsistent with "CBCs" in meaning. It is recommended to define the term as CBC (singular form) upon its first appearance (i.e., "chemical boundary condition (CBC)") and maintain consistent terminology throughout the manuscript.

2. Line 270: Although there are several definitions for the calculation formula of the Index of Agreement (IOA), the IOA formula presented in the manuscript differs from other established definitions. The authors are requested to verify and revise the IOA calculation formula accordingly, and to re-examine the IOA values reported in the paper.

3. On Line 404, the performance ranking of the dynamic CBC scenarios should be clarified. Please explicitly state that the ranking is based on all monitoring sites across China. Given that the NMB-based performance differs between avg-O3MDA8 and 90th-O3MDA8, separate rankings for these two metrics are recommended. Additionally, a discussion on the NMB performance across different regional subdivisions would strengthen this section.

4. Regarding Figure 4, the label "GEOSChem" is inconsistent with the text, which uses "GEOS-Chem". The figure should be revised to ensure consistency in nomenclature.

5. **On Section 3.2.3 and Section 2.2.2:** There is an inconsistency in the vertical data range described. The analysis in Section 3.2.3 is based on the 0-16 km range, whereas the description of the Vertical Observation Data in Section 2.2.2 states that all data were processed to 0-20 km. For consistency and clarity, the methodological description in Section 2.2.2 should be revised to reflect the 0-16 km range used in the subsequent analysis.

6. Table S1: According to the regional subregions description provided in the manuscript, the definition of the "Southwest China"—including its constituent provinces—is missing from Supplementary Table S1 and should be added to the table.

7. Table S4: There is a minor error in the table caption—a comma "," is missing between "IOA" and "r".